# Gender differences in the longitudinal association between obesity, and disability with workplace absenteeism in the Australian working population

**Syed Afroz Keramat**[1,2,3]*, **Khorshed Alam**[2,3], **Jeff Gow**[2,4], **Stuart J. H. Biddle**[3]

**1** Economics Discipline, Social Science School, Khulna University, Khulna, Bangladesh, **2** School of Commerce, University of Southern Queensland, Toowoomba, QLD, Australia, **3** Centre for Health Research, University of Southern Queensland, Toowoomba, QLD, Australia, **4** School of Accounting, Economics, and Finance, University of KwaZulu-Natal, Durban, South Africa

* syed.afroz@econ.ku.ac.bd

**Data Availability Statement:** This study requires no ethics approval for the authors as the analysis used only de-identified existing unit record data from the Household, Income and Labour Dynamics

## Abstract

### Background

Excess weight can increase absenteeism of workers and can have a negative influence on their productivity. Current evidence on this association is mostly based on cross-sectional data and there is little evidence concerning the longitudinal relationship between obesity, and disability with workplace absenteeism. Further, gender differences in this association have often ignored in the existing literature.

### Objectives

This study aims to examine gender differences in the longitudinal association between obesity, and disability with absenteeism in the workplace.

### Methods

Data from thirteen waves (2006 to 2018) of the Household, Income and Labour Dynamics in Australia (HILDA) survey were pooled, resulting in 117,769 observations for 19,851 adult employees. The Zero-Inflated Negative Binomial (ZINB) regression model was deployed to investigate the links between obesity, and disability with workplace absenteeism for the total sample and stratified by gender.

### Results

The findings showed that overweight (Incidence Rate Ratio [IRR]: 1.23, 95% confidence interval [CI]: 1.02–1.47), obesity (IRR: 1.35, 95% CI: 1.12–1.64) and disability (IRR: 2.83, 95% CI: 2.36–3.38) were associated with prolonged workplace absenteeism irrespective of gender. This study found that the multiplicative interaction between weight status and gender is significantly associated with absenteeism. The results reveal that the rate of absenteeism was 2.79 times (IRR: 2.79, 95% CI: 1.96–3.97) and 1.73 times (IRR: 1.73, 95% CI:

in Australia (HILDA) Survey. However, authors had completed and signed the Confidentiality Deed Poll and sent to NCLD (ncldresearch@dss.gov.au) and ADA (ada@anu.edu.au) before the data applications' approval. Therefore, datasets analysed and/or generated during the current study are subject to the signed confidentiality deed. The present study used The Household, Income and Labour Dynamics in Australia (HILDA) data set, which is a third party data set and were collected by the Melbourne Institute of Applied Economic and Social Research. There are some formalities on accessing and legal restrictions on sharing this data set. However, it should be mentioned here that the authors had not any special access privileges to the data set. Those interested in accessing this data should contact the Melbourne Institute of Applied Economic and Social Research, The University of Melbourne, VIC 3010, Australia, email: ncldresearch@dss.gov.au and ada@anu.edu.au.

**Funding:** This research did not receive any specific grant from funding agencies in the public, commercial or not-for-profit sectors.

**Competing interests:** The authors have declared that no competing interests exist.

**Abbreviations:** AUD, Australian Dollar; BMI, Body Mass Index; HILDA, Household, Income and Labour Dynamics in Australia survey; IRR, Incidence Rate Ratio; WHO, World Health Organization; ZINB, Zero-Inflated Negative Binomial.

1.20–2.48) higher among overweight and obese women than male counterparts, respectively. Moreover, this study found that the weight status of male workers is not associated with absenteeism. However, disability (IRR: 3.14, 95% CI: 2.43–4.05) is positively associated with longer days of absence among male workers. Finally, the study results showed that the rate of absenteeism is 1.82 (IRR: 1.82, 95% CI: 1.36–2.44), 1.61 (IRR: 1.61, 95% CI: 1.21–2.13), and 2.63 (IRR: 2.63, 95% CI: 1.99–3.48) times higher among overweight, obese, and female workers with a disability, respectively, compared with their lower weight counterparts.

## Conclusions

Workplace absenteeism is significantly associated with overweight and obesity among Australian workers. An active workplace health promotion program is very important for weight management of overweight and obese workers and thus to reduce workplace absenteeism. For example, employers may provide incentives for maintaining recommended body weights, encourage exercise, and promote healthy diets amongst their workers.

## Introduction

Globally, the prevalence of obesity has almost tripled since 1975 [1]. Worldwide more than 650 million adults aged 18 years or over were obese in 2016 [1]. Studies conducted on US workers provide evidence that obese employees were more likely to be absent from the workplace compared to their healthy weight counterparts [2–4]. Moreover, a study in Ireland concludes that obese employees were 72% more prone to be absent [5]. Further, a recent study in the Netherlands revealed that obese workers took 14 days of extra leave per annum compared to their lower weight counterparts [6]. Similar results have been found in a British study where the authors claimed that obese workers were absent for four extra days per year [7]. However, a study in Germany did not find evidence that overweight men took more sick leave days [8]. A few studies have also examined the longitudinal association between obesity and workplace absenteeism [9–12]. A prospective study among middle-aged employees in Finland revealed that stable obesity and weight gain in the follow-up period increased the risk of prolonged sickness absence [12]. Two US-based longitudinal studies also provided evidence that obesity is positively associated with absenteeism [10,11].

The prevalence of overweight and obesity among Australian adults is 63% and its rising prevalence has become a serious public health concern [13]. The high health and financial burden of overweight and obesity in Australia has been well documented [14]. Excess weight in individuals is responsible for 7% of the total health burden in the country [14]. The direct financial cost of obesity to the Australian economy was estimated to be AUD 3.8 billion in 2011–12 [15]. In addition to the direct costs, overweight and obesity have indirect costs in the form of lost productivity (i.e. increased absenteeism and presenteeism). In 2011–12, the indirect cost of obesity was estimated to be AUD 4.8 billion to the Australian economy [15].

Absenteeism in the Australian workplace has risen by 7% since 2010 [16]. Approximately ninety-two million workdays are lost annually with the annual cost in the form of lost productivity is estimated to be AUD 33 billion [16]. This is up to 8% of the total payroll costs to Australian companies [16]. The main reasons for employees' absence are poor health and fitness [17], illness (flu, headache, and gastro), family responsibilities, mental issues, and alcohol/

drug-related issues [16]. Employees who are absent from the workplace due to personal illness or injury include obese individuals who take longer leave periods compared with non-obese individuals [18]. According to the National Health Survey (NHS), over 4 million workdays were lost from Australian workplaces in 2001 due to obesity [18]. This evidence suggests that there might be an association between weight status and absenteeism for the Australian working population.

A few studies that have attempted to identify the longitudinal relationship between obesity and workplace absenteeism have mostly been based in the US or European countries. Evidence on the relationship between obesity and disability with workplace absenteeism from the Australian perspective is still lacking. Additionally, very few studies have investigated gender differences in the longitudinal association between obesity, disability, and workplace absenteeism. The present study fills this void in the literature by addressing the research question: does gender difference exist in the longitudinal association between obesity and disability with absenteeism in the workplace?

Excessive bodyweight of workers should be a major concern to businesses as there might be a positive association between workplace absenteeism and obesity and thus the extra cost to companies. The present study will offer evidence on the longitudinal links between obesity, disability, and workplace absenteeism. The results of the study might be used by policymakers and organizations for the development and implementation of workplace health promotion programs to tackle excessive weight problems of the workers.

## Materials and methods

### Data source and sample selection

The present study used the individual person dataset from the Household, Income and Labour Dynamics in Australia (HILDA) survey. This is a large-scale nationally representative panel survey of Australian households that collects data on family, wealth, health, education, and labor market dynamics [19]. This household panel survey is similar to the Panel Study of Income Dynamics (PSID) in the US, the British Household Panel Survey (BHPS), and the German Socio-Economic Panel (SOEP). The HILDA survey commenced in 2001 and since then has been conducted annually following the University of Melbourne's ethical guidelines. It collects detailed information from household members aged 15 years and over using a combination of face-to-face interviews and telephone interviews by trained interviewers, and self-completed questionnaires. There is a concern that responses collected through different modes have a significant impact on data quality. However, preliminary findings suggest that there is little systematic variation in responses by data collection modes [20].

This study utilized twelve recent waves (waves 6 to 18) from the HILDA dataset. The main reason for choosing the most recent 13 waves of the survey (2006–2018) is that data on Body Mass Index (BMI) are available only in these waves. The inclusion criteria of the present study are participants aged 15–64 years and who are employed at each wave. Missing information on the outcome variable of days absent from the workplace in the last 12 months were excluded (n = 2368 observations). Further, pregnant female employees were excluded (n = 6364 observations) from the subsample analyses to avoid potential bias and ensuring the validity of the study findings. After employing inclusion criteria and excluding missing data, the unbalanced panel consists of 117,769 observations from 19,851 adult employees. Study participants were generated from the dataset following the HILDA survey protocol. HILDA uses a multi-stage sampling approach including sampling within households within a particular administrative area. Detailed information about the sampling procedure and design have been described elsewhere [19].

The percentage of participants who were lost due to missing information on the outcome variable and to pregnancy was 2.01% and 5.40%, respectively. The total percentage of loss to follow-up in the present study is less than 10%. That is in the acceptable range for longitudinal studies and thus leads to little bias.

## Measures

**Outcome variable.** The main outcome variable of the study is days absent from work on paid workers' compensation in the last twelve months. It is a derived variable and was constructed using the variable work schedule to determine the number of days absent from the workplace.

**Gender differences.** Work and health-related behaviors often differ by gender [21]. The existing evidence reported mixed results when explaining the association between obesity and absenteeism [7]. The inconsistent findings may be due to variables that moderate the relationship. Previous studies identified the variable, gender, which moderates the association between job-related factors and workplace absenteeism [22]. Attendance rate is an avenue by which women differ from men at the workplace [23]. Keeping this in mind, the present study conducts gender-specific analyses while examining the longitudinal association between obesity, disability, and absenteeism. Moreover, this study will include a multiplicative interaction term, BMI × gender, in the regression model to test whether the joint effect of BMI and gender is significant in explaining workplace absenteeism.

**Exposure variables.** The main variables of interest in the present study are BMI and disability. BMI is calculated using self-reported height and weight following the formula weight (in kilograms) divided by height (in meters squared). This study categorized BMI into four groups following the World Health Organization (WHO) guidelines: <18.50 (underweight), 18.50–24.99 (normal/healthy weight), 25.00–29.99 (overweight), and ≥30.00 (obesity) [1]. The obesity often further categorized into three groups: 30.00–34.99 (obese class I), 35.00–39.99 (obese class II), and ≥40.00 (obese class III). Underweight is not a topic of interest in the current study. As a result, this study merged two BMI categories (underweight with healthy weight) and form a new category, <25 BMI, following relevant studies conducted in Australia and The Netherlands [24,25] to conduct the regression analysis.

The disability of an adult used in the HILDA survey was based on the guidelines of the International Classification of Functioning, Disability, and Health (ICF) under the WHO framework [26]. Participants were asked if they have any 'disability, impairment, or disability that restricts them in everyday activities, and has lasted or are likely to last, for 6 months or more' [27]. Responses were coded in binary form (yes or no). Participants who answered 'yes' were counted as an adult with a disability.

**Other covariates.** This study selected potential confounders following relevant published studies on the risk factors of workplace absenteeism [3–12,25,28,29] and information available in the HILDA datasets. Confounders were included in the fully adjusted model only if a confounder was found significant at 5% or less risk level at any level in the bivariate analyses.

This study includes age (15–25, 26–45, 46–60, and over 60 years) [2,4,7,9], gender (male and female) [4,7,10], civil status (non-cohabiting and married/cohabiting) [9], and education (year 12 or below, professional qualification, and university qualification) [2,4] as socio-demographic characteristics.

The present study also included eleven measures of job-related characteristics that include firm size (small, medium, and large) [30], employment contract (permanent, fixed-term and casual) [30,31], tenure with the current employer (1–5 years, and 6 or more years) [30], hours worked per week (<35, 35–40, and >40 hours a week) [32], work schedule (day and shift

work) [30,32], job type (non-manual and manual) [4,32], supervisory responsibility (yes and no), paid holiday leave (yes and no), paid sick leave (yes and no) [11,32], union membership (yes and no) [30,31], and overall job satisfaction (dissatisfied, neutral, and satisfied)[32].

Confounding role of other comorbidities such as cancer, diabetes, heart disease, depression, asthma, bronchitis, and arthritis in explaining workplace absenteeism could not be explored in the present study. The principal reason for not exploring such roles is that these data were available only in waves 9, 13, and 17 of the HILDA survey.

**Estimation strategy.** The authors constructed an unbalanced longitudinal data set consisting of 117,769 observations by linking 19,851 individuals' records who participated in either any of the waves from 6 to 18 of the HILDA survey. Descriptive statistics in the form of frequency (n) and percentages (%) with 95% confidence intervals (CI) or mean (SD) or median (range) were used to describe absenteeism, weight status, disability, socio-demographic and job-related characteristics of the study participants.

To explore the factors associated with workplace absenteeism, the present study followed the conceptual framework of Hafner et al. [33]. Accordingly, factors of workplace productivity (absenteeism and presenteeism) are broadly categorized into three groups and can be expressed as follows.

$$Y_i = f(j, \ p, \ h)$$

In the function, $Y_i$ refers to workplace productivity (i.e. absenteeism), j refers to job-related factors (i.e. work demands), p refers to personal factors (i.e. lifestyle factors), and h refers to health and physical factors (i.e. long-term health conditions).

To find out the longitudinal association between exposure and outcome variables, the present study followed the forward addition approach for building models. In this approach, the multivariate model starts with the basic model where BMI is the exposure, and absenteeism is the outcome variable. Confounders and interaction terms were added one at a time based on their level of significance. The process continued until all significant confounders and interaction term was included in the model.

The outcome variable, workplace absenteeism, is a count variable where all the values are non-negative integer numbers including zero. The negative binomial model is appropriate to estimate the association between exposures and the outcome variable when the outcome variable is a count variable and overdispersed [34]. In the present study, the number of zeros in the outcome variable is excessive. Among these zeros, there are two kinds of zero values. First, there are some certain zeros because employees may not be absent in the workplace due to work restrictions. Second, there might exist zeros for employees who were not absent in the workplace but could be absent due to sickness or other conditions. Hence, the number of zeros might be inflated in the outcome variable due to certain zeroes. The standard negative binomial regression model cannot differentiate between these two processes when they arrive at a zero value in the outcome variable [35]. However, the Zero-Inflated Negative Binomial (ZINB) model can handle these two distinct data generation processes [35]. The ZINB model fits a logistic regression model to predict the excess zeros in the dependent variable (absenteeism) and then fits the negative binomial regression model to get a count of the number of days absent for non-excess zeros [36]. Given this, the current study followed standard practice and employed the ZINB regression model to estimate the longitudinal association between obesity, disability, and workplace absenteeism. The study results are demonstrated in the form of the incidence rate ratio (IRR) for each variable. Stata 14 windows version was used for all statistical analyses. This study set a p-value at <0.05 level for statistical significance.

**Ethics approval.** This study requires no ethics approval for the authors as the analysis used only de-identified existing unit record data from the Household, Income and Labour Dynamics in Australia (HILDA) Survey. However, the authors had completed and signed the Confidentiality Deed Poll and sent it to NCLD (ncldresearch@dss.gov.au) and ADA (ada@anu.edu.au) before the data applications' approval. Therefore, datasets analyzed and/or generated during the current study are subject to the signed confidentiality deed.

## Results

### Descriptive characteristics of the study sample

Table 1 shows the pooled characteristics of the employees in terms of overweight, obesity, disability, absenteeism, socio-demographic characteristics, and job-related characteristics. Among the study participants, around 52% were either normal weight or underweight (<25 BMI), 29% were overweight, and 19% were obese. An estimated 16% of Australian workers have a disability. The average number of absent days per annum of workers is 0.7, although the standard deviation (8.8) is very high. A higher value of the standard deviation over mean indicates the absent days variable is overdispersed with excessive zeros. Additionally, Table 1 reports that median absent days of the employees is 0.00 and ranges from 0 to 352 days.

Fig 1 demonstrates that average absenteeism is significantly higher among overweight and obese employees compared with lower weight employees. Fig 1 illustrates that the average number of missed days is highest among the morbidly obese (obese class III) workers (1.79), followed by workers belong to obese class II (1.23 days).

### Factors associated with workplace absenteeism

Estimates of the longitudinal association between obesity, and disability with absenteeism after controlling for socio-demographic and job-related characteristics are presented in Table 2.

The results showed a set of significant links between overweight, obesity, and disability with absenteeism in the adjusted model (model 1). The results showed that overweight, obesity, and disability have a longitudinal association with absenteeism. The findings indicate that the rate of workplace absenteeism in overweight and obese workers were 1.23 (IRR: 1.23, 95% CI: 1.02–1.47) and 1.35 (IRR: 1.35, 95% CI: 1.12–1.64) times higher compared with their lower weight counterparts, respectively. Model 1 also reveals that the rate of days absent from the workplace among workers with a disability was 2.83 times (IRR: 2.83, 95% CI: 2.36–3.38) higher compared with workers without a disability. Model 2 reports a significant association between the interaction of BMI and gender with prolonged absenteeism. The results showed that the rate of absenteeism was 2.79 times (IRR: 2.79, 95% CI: 1.96–3.97) and 1.73 times (IRR: 1.73, 95% CI: 1.20–2.48) higher among overweight and obese women employees than their male counterparts, respectively.

The present study also explored the relationship between obesity, disability, with absenteeism by gender. Model 3 and Model 4 report the results obtained from multivariate models for male and female workers, respectively. The adjusted model (model 3) showed that male workers' weight status is not associated with workplace absenteeism. However, the study findings suggest that the rate of absenteeism in male workers with a disability is 3.14 times (IRR: 3.14, 95% CI: 2.43–4.05) higher compared with lower weight counterparts. Model 4 shows that there is a longitudinal association between female workers' weight status, disability with absenteeism. After adjusting confounders, model 4 also reveals that the rate of absenteeism among overweight and obese women workers were 1.82 (IRR: 1.82, 95% CI: 1.36–2.44) and 1.61 (IRR: 1.61, 95% CI: 1.21–2.13) times higher compared with lower weight peers, respectively. The

**Table 1. Background characteristics of the study participants.**

| Variables | N | % (95% CI) |
|---|---|---|
| Outcome Variable: Days absent in the past 12 months (mean [SD]) | 117,769 | 0.7 (8.8) 32.9 (50.9) without counting 0 days (median = 0.0; min = 0, max = 352) |
| Explanatory variables | | |
| Health-related characteristics | | |
| BMI | | |
| *BMI (<25)* | 61,102 | 51.9 (51.6–52.2) |
| *Overweight (25.00–29.99)* | 34,532 | 29.3 (29.1–29.6) |
| Obesity (≥30.00) | 22,135 | 18.8 (18.6–19.1) |
| *Obese class I (30.00–34.99)* | 14,749 | 12.5 (12.3–12.7) |
| *Obese class II (35.00–39.99)* | 5,052 | 4.3 (4.2–4.4) |
| *Obese class III (≥40.00)* | 2,334 | 2.0 (1.9–2.1) |
| Disability | | |
| *No* | 98,477 | 83.6 (83.4–83.8) |
| *Yes* | 19,292 | 16.4 (16.2–16.6) |
| Socio-demographic characteristics | | |
| Age | | |
| *15–25 years* | 25,960 | 22.1 (21.8–22.3) |
| *26–45 years* | 49,867 | 42.3 (42.1–42.6) |
| *46–60 years* | 37,011 | 31.4 (31.2–31.7) |
| *>60 years* | 4,931 | 4.2 (4.1–4.3) |
| Gender | | |
| *Male* | 60,204 | 51.1 (50.8–51.4) |
| *Female* | 57,565 | 48.9 (48.6–49.2) |
| Civil status | | |
| *Non-Cohabiting* | 46,884 | 39.8 (39.5–40.0) |
| *Married/Cohabitating* | 70,885 | 60.2 (59.9–60.5) |
| Education | | |
| *Year 12 or below* | 44,421 | 37.7 (37.4–38.0) |
| *Professional qualification* | 39,369 | 33.4 (33.2–33.7) |
| *University qualification* | 33,979 | 28.9 (28.6–29.1) |
| Job-related characteristics | | |
| Farm size | | |
| *Small (1–19 employees)* | 51,704 | 43.9 (43.6–44.2) |
| *Medium (20–99 employees)* | 32,314 | 27.4 (27.2–27.7) |
| *Large (≥100 employees)* | 33,751 | 28.7 (28.4–28.9) |
| Employment contract | | |
| *Permanent* | 78,442 | 66.6 (66.3–66.9) |
| *Fixed-term* | 11,600 | 9.9 (9.7–10.0) |
| *Casual* | 27,727 | 23.5 (23.3–23.8) |
| Tenure-current employer | | |
| *1–5 years* | 65,326 | 55.5 (55.2–55.8) |
| *6 or more years* | 52,443 | 44.5 (44.2–44.8) |
| Hours worked per week | | |
| *<35 hours/week* | 37,836 | 32.1 (31.9–32.4) |
| *35–40 hours/week* | 42,432 | 36.1 (35.8–36.3) |
| *>40 hours/week* | 37,501 | 31.8 (31.6–32.1) |
| Work schedule | | |

*(Continued)*

**Table 1.** (Continued)

| Variables | N | % (95% CI) |
|---|---|---|
| *Day work* | 88,769 | 75.4 (75.1–75.6) |
| *Shift work* | 29,000 | 24.6 (24.4–24.9) |
| Job type | | |
| *Non-manual* | 59,582 | 50.6 (50.3–50.9) |
| *Manual* | 58,187 | 49.4 (49.1–49.7) |
| Supervisory responsibilities | | |
| *Yes* | 53,490 | 45.4 (45.1–45.7) |
| *No* | 64,279 | 54.6 (54.3–54.9) |
| Paid holiday leave | | |
| *Yes* | 85,447 | 72.5 (72.3–72.8) |
| *No* | 32,322 | 27.5 (27.2–27.7) |
| Paid sick leave | | |
| *Yes* | 85,709 | 72.8 (72.5–73.0) |
| *No* | 32,060 | 27.2 (27.0–27.5) |
| Union membership | | |
| *Yes* | 26,967 | 22.9 (22.7–23.1) |
| *No* | 90,802 | 77.1 (76.9–77.3) |
| Overall job satisfaction | | |
| *Dissatisfied* | 3,006 | 2.6 (2.5–2.7) |
| *Neutral* | 17,649 | 15.0 (14.8–15.2) |
| *Satisfied* | 97,114 | 82.4 (82.2–82.7) |

Abbreviations: SD Standard Deviation; CI Confidence Interval

present study also showed that the rate of absenteeism among women with disabilities is 2.63 times (IRR: 2.63, 95% CI: 1.99–3.48) higher than women without a disability.

## Discussion

The purpose of the present study is to assess the longitudinal association between obesity, and disability with workplace absenteeism in Australian workers, and to test for gender differences in such associations. This study pooled 13 waves of data from the nationally representative sample of the HILDA survey. Controlling for socio-demographic and job-related characteristics, ZINB regression analysis showed that overweight and obesity are associated with prolonged absenteeism for the entire sample. Some observational studies also confirm that obese workers tend to have a higher number of work absences [2,4–7,28,29]. In addition to cross-sectional study findings in the literature, a recent study has also confirmed a longitudinal association between obesity and workplace absenteeism [11]. It was already well documented that obesity is a major risk factor for many chronic diseases [1]. Obese workers missed more days of work due to personal illness or injury compared with non-obese workers [18]. Further, the present study revealed that having a disability is significantly associated with prolonged absenteeism irrespective of gender. This finding is in line with a study from the Netherlands where the authors found that long-term health condition like distress is positively associated with long-term sickness absence [25]. The association between disability and higher absenteeism might be explained by the fact that comorbidities lead to a higher number of absent days [6,25].

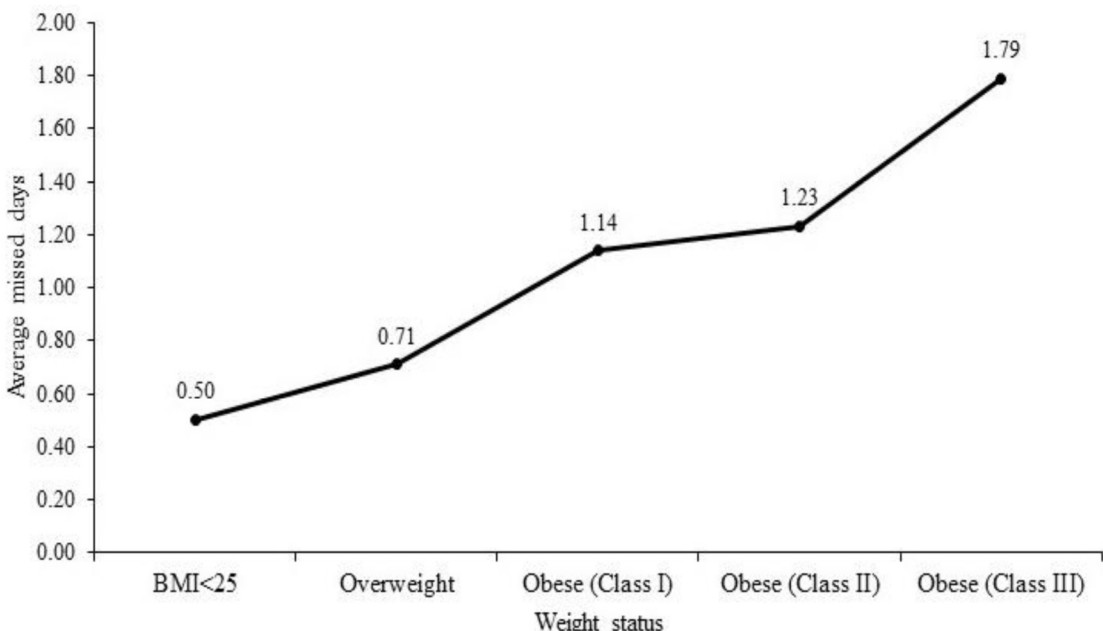

**Fig 1. Average number of missed days according to weight status.**

The present study also found a significant multiplicative interaction of BMI and gender in explaining workplace absenteeism. The study results revealed that the rate of absenteeism is higher among overweight and obese women than male counterparts. Additionally, the present study checks the longitudinal association between BMI and prolonged absenteeism separately for male and female workers. The current study results showed that there is no longitudinal association between overweight, obesity, and a high rate of absenteeism among male workers. However, the results found that overweight, obesity, and absenteeism are positively associated in the long-run among female workers. An existing longitudinal study supports the present study findings as it found obesity was associated with extra sick leave days and long-term workplace absenteeism in female but not in male workers [9]. An important cause of this gender difference in workplace absenteeism may be the menstrual cycle [37]. Further, the gender difference in absenteeism could be attributed to women's double burden of wage work and unpaid household chores [38]. Another possible explanation is that women typically perform more monotonous and stressful jobs [38].

Knowledge of the longitudinal association between obesity and absenteeism is important to companies and policymakers to take measures to reduce the rate of absenteeism in the workplace [9]. From the viewpoint of public policy, the results of this longitudinal study will help policymakers to have a more comprehensive understanding of absenteeism in the workplace due to excessive weight. The results suggest that organizations should focus on an integrated lifestyle approach for weight management of their workers by using multiple intervention strategies. Organizations should create a supportive environment by enabling physical infrastructure and workplace culture to encourage a healthy lifestyle. For example, companies may offer healthy catering services, establish gym and activity centers for physical activity, establish on-site bicycle storage, and provide walking maps and routes. The effectiveness of workplace-targeted interventions is currently unclear. However, there is evidence that the absenteeism rate is low among workers who perform physical activities regularly [39,40].

**Table 2. ZINB regression results for factors associated with workplace absenteeism[a].**

| Variables | Model 1 (total sample) IRR (95% CI)[b] | Model 2 (total sample) IRR (95% CI)[c] | Model 3 (only male) IRR (95% CI)[d] | Model 4 (only female) IRR (95% CI)[e] |
|---|---|---|---|---|
| BMI | | | | |
| *BMI (<25) (ref)* | | | | |
| *Overweight (25.00–29.99)* | **1.23 (1.02–1.47)** | | 0.96 (0.76–1.22) | **1.82 (1.36–2.44)** |
| *Obesity (≥30.00)* | **1.35 (1.12–1.64)** | | 1.26 (0.96–1.65) | **1.61 (1.21–2.13)** |
| Disability | | | | |
| *No (ref)* | | | | |
| *Yes* | **2.83 (2.36–3.38)** | **2.89 (2.42–3.46)** | **3.14 (2.43–4.05)** | **2.63 (1.99–3.48)** |
| Gender | | | | |
| *Male (ref)* | | | | |
| *Female* | 0.97 (0.81–1.16) | | | |
| Interaction terms (BMI × *Gender*) *Male × BMI (<25) (ref)* | | | | |
| *Overweight × female* | | **2.79 (1.96–3.97)** | | |
| *Obesity × female* | | **1.73 (1.20–2.48)** | | |
| Socio-demographic characteristics | | | | |
| Age | | | | |
| *15–25 years (ref)* | | | | |
| *26–45 years* | **1.47 (1.19–1.83)** | **1.52 (1.23–1.88)** | 1.11 (0.82–1.49) | **2.06 (1.47–2.88)** |
| *46–60 years* | **1.81 (1.43–2.29)** | **1.93 (1.53–2.44)** | **1.47 (1.06–2.04)** | **2.56 (1.77–3.69)** |
| *>60 years* | **1.67 (1.09–2.56)** | **1.70 (1.11–2.61)** | 0.78 (0.42–1.44) | **2.79 (1.43–5.42)** |
| Civil status | | | | |
| *Non-Cohabitating (ref)* | | | | |
| *Married/Cohabitating* | 0.90 (0.77–1.05) | 0.94 (0.80–1.09) | 1.03 (0.83–1.29) | 1.00 (0.79–1.26) |
| Education | | | | |
| *Year 12 or below* | **1.75 (1.38–2.22)** | **1.76 (1.39–2.22)** | **3.64 (2.60–5.11)** | 0.96 (0.71–1.31) |
| *Professional qualification* | **1.92 (1.51–2.43)** | **1.93 (1.52–2.44)** | **3.50 (2.50–4.91)** | 0.89 (0.67–1.19) |
| *University qualification (ref)* | | | | |
| Job-related characteristics | | | | |
| Farm size | | | | |
| *Small (1–19 employees)* | 1.09 (0.90–1.31) | 1.07 (0.89–1.30) | 1.10 (0.85–1.43) | 0.96 (0.71–1.31) |
| *Medium (20–99 employees)* | 1.00 (0.83–1.21) | 0.97 (0.81–1.18) | 1.00 (0.77–1.30) | 0.89 (0.66–1.19) |
| *Large (≥100 employees) (ref)* | | | | |
| Employment contract | | | | |
| *Permanent (ref)* | | | | |
| *Fixed-term* | 0.88 (0.68–1.14) | 0.85 (0.66–1.10) | 0.93 (0.65–1.33) | 0.77 (0.52–1.15) |
| *Casual* | 0.84 (0.58–1.22) | 0.78 (0.54–1.12) | 0.73 (0.46–1.17) | 1.38 (0.70–2.75) |
| Tenure-current employer | | | | |
| *1–5 years (ref)* | | | | |
| *6 or more years* | 0.86 (0.73–1.01) | **0.82 (0.69–0.96)** | **0.76 (0.61–0.96)** | 0.91 (0.71–1.18) |
| Hours worked per week | | | | |
| *<35 hours/week* | **0.80 (0.66–0.99)** | **0.79 (0.65–0.97)** | **0.72 (0.52–0.99)** | 0.81 (0.61–1.07) |
| *35–40 hours/week (ref)* | | | | |
| *>40 hours/week* | 0.99 (0.83–1.18) | 0.98 (0.82–1.16) | 0.95 (0.77–1.17) | 0.85 (0.60–1.19) |
| Work schedule | | | | |
| *Day work (ref)* | | | | |
| *Shift work* | 1.18 (0.99–1.40) | **1.21 (1.02–1.43)** | **1.51 (1.19–1.91)** | 1.20 (0.91–1.57) |
| Job type | | | | |

*(Continued)*

**Table 2.** (*Continued*)

| Variables | Model 1 (total sample) IRR (95% CI)[b] | Model 2 (total sample) IRR (95% CI)[c] | Model 3 (only male) IRR (95% CI)[d] | Model 4 (only female) IRR (95% CI)[e] |
|---|---|---|---|---|
| *Non-manual (ref)* | | | | |
| *Manual* | **2.00 (1.63–2.48)** | **2.03 (1.66–2.50)** | **2.55 (1.94–3.35)** | **1.61 (1.17–2.21)** |
| Supervisory responsibilities | | | | |
| *Yes (ref)* | | | | |
| *No* | 0.93 (0.80–1.08) | 0.93 (0.81–1.08) | 0.97 (0.79–1.19) | 0.85 (0.68–1.08) |
| Paid holiday leave | | | | |
| *Yes (ref)* | | | | |
| *No* | 0.96 (0.43–2.15) | 1.10 (0.49–2.45) | 1.21 (0.46–3.17) | 0.59 (0.18–1.98) |
| Paid sick leave | | | | |
| *Yes (ref)* | | | | |
| *No* | 0.87 (0.38–1.99) | 0.79 (0.34–1.81) | 0.66 (0.24–1.83) | 1.01 (0.29–3.58) |
| Union membership | | | | |
| *Yes (ref)* | | | | |
| *No* | **0.53 (0.43–0.66)** | **0.55 (0.43–0.66)** | **0.60 (0.46–0.78)** | **0.51 (0.36–0.72)** |
| Overall job satisfaction | | | | |
| *Dissatisfied* | **1.57 (1.07–2.31)** | **1.54 (1.06–2.25)** | 1.60 (0.93–2.75) | 1.46 (0.83–2.54) |
| *Neutral* | 1.15 (0.95–1.41) | 1.21 (0.99–1.48) | 0.78 (0.60–1.02) | **1.70 (1.25–2.33)** |
| *Satisfied (ref)* | | | | |

Abbreviations: BMI Body Mass Index; CI Confidence Interval; IRR Incidence Rate Ratio; Ref Reference

[a]Values in bold are statistically significant at p<0.05

[b]Estimates of obesity and disability after adjusting socio-demographic and job-related characteristics using the total sample (model 1)

[c]Estimates of the interaction of BMI and gender using the total sample (model 2).

[d]Estimtes of obesity and disability after adjusting socio-demographic and job-related characteristics using male samples only (model 3).

[e]Estimtes of obesity and disability after adjusting socio-demographic and job-related characteristics using female samples only (model 4).

The study contributes to the existing literature in several ways. First, to the best of the author's knowledge, this is the first study on the longitudinal association between obesity, disability, and absenteeism from the Australian context. Second, the present study pooled a nationally representative longitudinal sample of 117,769 observations for 19,851 workers where participants were observed for 13 years to offer precise estimates on the association. Third, the study incorporated a large number of job-related characteristics as confounders including less investigated factors (work schedule, job type, paid, and sick leave arrangement) which are associated with absenteeism. Fourth, this is the first study that examines the effect of the interactions between BMI and gender on absenteeism.

## Conclusions

This study aimed to examine the gender differences in the longitudinal association between obesity, and disability with absenteeism. Using the ZINB regression technique, the present study found evidence of significant association and compared the results with existing evidence. The study found that workplace absenteeism is higher among overweight, obese, and workers with a disability compared with their counterparts. The results also revealed that interactions of BMI and gender are associated with prolonged absenteeism. This study found evidence that the rate of absenteeism is higher among overweight and obese women than male counterparts. However, the study results did not find evidence of a longitudinal association between overweight, and obesity with a high rate of absenteeism among male workers. The

findings are important evidence in the consideration of workplace health promotion policies. Implementation of workplace health promotion programs to treat workers excess weight might be an effective tool to lower the rate of absenteeism.

The present study has some limitations. First, the unbalanced longitudinal design of the study draws longitudinal associations but it is not possible to discern the causal effect of obesity, and disability on workplace absenteeism. Second, the study findings might be vulnerable to bias, as data on BMI, disability, and absenteeism are self-reported. Self-reported bias is high among overweight and obese adults, as they tend to overestimate their height and underestimate their weight [41,42]. Similarly, there might be justification bias in case of self-reported disability as individuals tended to over-report their disability level as a result of the financial benefits attached to that classification [43]. The authors call for a well-designed cohort study that can draw causal inferences on the association between obesity, disability, and absenteeism.

## Acknowledgments

The authors would like to thank the Melbourne Institute of Applied Economic and Social Research for providing the HILDA data set.

## Author Contributions

**Conceptualization:** Syed Afroz Keramat.

**Data curation:** Syed Afroz Keramat.

**Formal analysis:** Syed Afroz Keramat.

**Methodology:** Syed Afroz Keramat.

**Software:** Syed Afroz Keramat.

**Supervision:** Khorshed Alam, Jeff Gow, Stuart J. H. Biddle.

**Writing – original draft:** Syed Afroz Keramat, Stuart J. H. Biddle.

**Writing – review & editing:** Khorshed Alam, Jeff Gow.

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
