## [Decision Letter · Decision Letter 0]

20 Feb 2020

PONE-D-19-32153

Gender differences in the longitudinal association between obesity, disability, and workplace absenteeism in the Australian working population

PLOS ONE

Dear authors,

Thank you for submitting your manuscript to PLOS ONE. After careful consideration, we feel that it has merit but does not fully meet PLOS ONE’s publication criteria as it currently stands. Therefore, we invite you to submit a revised version of the manuscript that addresses the points raised during the review process.

1. Please rearrange the introduction (attached document)

2. Some results are discussed in the results section which should go to the discussion. Some statements related to materials and methods are in the results section. Some results are presented with too details or too descriptive whereas those are presented in Tables and Figures. I would consider to present the key findings and refer the rest to the Tables/Figures.

3. Absenteeism is too skewed. Please present with median and range.

4. Although gender differences are the major focus, it was not presented in the methods section.

5. Please report what are the models 1, 2, 3 and 4 are (covariate-adjusted) and present only Model 1 and Model 2. If there are no changes in models after additional covariate adjustment, no points to show the Models. It can be said in the text that this………….was done but the model did not change. Even, Model 1 can be presented in the text and Model 2 can be presented in Table 2 for Total and by Gender by combining Table 2, 3, and 4. This will give side by side comparison without losing any information.

We would appreciate receiving your revised manuscript by Apr 05 2020 11:59PM. To enhance the reproducibility of your results, we recommend that if applicable you deposit your laboratory protocols in protocols.io, where a protocol can be assigned its own identifier (DOI) such that it can be cited independently in the future. For instructions see: http://journals.plos.org/plosone/s/submission-guidelines#loc-laboratory-protocols

We look forward to receiving your revised manuscript.

Kind regards,

Fakir M Amirul Islam, PhD

Academic Editor

PLOS ONE

Journal Requirements:

2. In ethics statement in the manuscript and in the online submission form, please provide additional information about the patient records used in your retrospective study. Specifically, please ensure that you have discussed whether all data were fully anonymized before you accessed them and/or whether the IRB or ethics committee waived the requirement for informed consent. If patients provided informed written consent to have data from their medical records used in research, please include this information.

4. Please include your tables as part of your main manuscript and remove the individual files. Please note that supplementary tables (should remain/ be uploaded) as separate "supporting information" files

Reviewers' comments:

Reviewer's Responses to Questions

**Comments to the Author**

1. Is the manuscript technically sound, and do the data support the conclusions?

Reviewer #1: Partly

2. Has the statistical analysis been performed appropriately and rigorously? 

Reviewer #1: No

3. Have the authors made all data underlying the findings in their manuscript fully available?

Reviewer #1: Yes

4. Is the manuscript presented in an intelligible fashion and written in standard English?

Reviewer #1: No

5. Review Comments to the Author

Reviewer #1: Thank you for inviting me for reviewing this manuscript.

The authors conducted a longitudinal study for investigating the gender differences in the association between obesity, disability, and workplace absenteeism in the Australian working population. A large recruited sample size as well as its longitudinal nature are the strength of this study. However, there are a number of concerns threatening the validity of the study results.

Abstract

Please don’t use the abbreviation when using IRR for the first time.

Line 41: Please clearly determine which type of interaction i.e. additive or multiplicative has not been significant?

Line 44: Conclusions: what about male workers?

INTRODUCTION1

Generally, the introduction section is too long to follow (more than 900 words) and needs reducing the number of utilized words.

METHOD

Line 128-130: The different employed method for obtaining the study information i.e. a combination of personal face-to-face interviews by trained interviewers and self-completed questionnaires might impose an information bias to the study findings. Please clarify this discrepancy.

Line 135-136: what do you mind by “prospective subjects” in the following, sentences: “The inclusion criteria of the study are prospective subjects aged 15-64 years and employed at each wave.”

Line 138: please clarify for your readers why pregnant employees with a large sample size (n = 8631 observations) were excluded from the subsample analyses.

Line 148-149: Although the authors stated the self-reported nature of BMI in the limitation section, however, the self-reported BMI as well as disability could imposed an important information bias to the study finding.

Line 164: what was the utilized criteria for selecting potential confounder?

Line 166: What about the role of sex (male and female)? Is it confounder or effect modifier?

There is not information about the potential confounding role of the other comorbidities.

Please explain the goodness of fit criteria and model-fitting process for 4 introduced models.

The results show that obese class I and II were significantly associated with absenteeism but not with overweight and obese class III. Please explain this inconsistent finding?

Please revise the reported interpretation of IRR throughout the manuscript. Please note that the RATE measures the rate of the occurrence of the study event (here, absenteeism) not its probability. For example when interpreting the rate of obese class II workers (2.18 (IRR: 2.18, 95% CI: 1.43-3.33)) with their lower weight counterparts, the IRR tells the rate of absenteeism in obese class II workers compared with the lower weight. (Similar interpretation has reported in lines: 256, 257, …)

Line 258: please state which type of interaction; additive or multiplicative has not been identified between obesity and disability and prolonged absenteeism.

Please clarify the sample size and its possible over-power.

Please test the possible interaction between sex and overweight in absenteeism.

Table 1

What is 8.70 in the first row? Is it SD? If yes, please explain why it is very larger than its Mean? Please separately report the mean (SD) in non-zeros.

While obese class II has shown a significant association with absenteeism, obese class III has not shown such an association. Please explain?

Please use “confounder” throughout the manuscript instead of control variables.

Obese class III (≥40.00) has shown a significant association only in model I but not in the other models. Please explain.

Please test and report any significant trend for BMI variable.

Table 4

An unexpected point estimates of potential interactions in table 4 is reported for possible interaction between Obesity and disability. The reported point estimates are protective! Please explain.

Some reported numbers in Table 4 is without decimal numbers. Please keep a unique scheme for representing the numbers.

The point estimates for interaction terms in Table 3 are surprising. While the interaction terms between Obese class I and having disability is larger than 1, this is protective for Obese class II and having disability!

There is no information regarding the percent of lost to follow up and its potential impact on the study findings.

Line 300: Please explain the reported discrepancy between the significant interactions between the study variables in male but not females?

CONCLUSSION

With the study limitations as well as

6. PLOS authors have the option to publish the peer review history of their article (what does this mean?). If published, this will include your full peer review and any attached files.

Reviewer #1: No

---

## [Author Response · Author response to Decision Letter 0]

14 Apr 2020

Thank you for allowing us to revise our manuscript entitled “Gender differences in the longitudinal association between obesity, disability, and workplace absenteeism in the Australian working population”. We have found the reviewers’ comments/feedback very helpful in improving the manuscript and we have revised the manuscript accordingly. Additionally, we have addressed the journal requirements. Please find attached the revised manuscript.

---

## [Editor Report · Decision Letter 1]

4 May 2020

PONE-D-19-32153R1

Gender differences in the longitudinal association between obesity, disability, and workplace absenteeism in the Australian working population

PLOS ONE

Dear Ms Karamat,

Thank you for submitting your manuscript to PLOS ONE. After careful consideration, we feel that it has merit but does not fully meet PLOS ONE’s publication criteria as it currently stands. Therefore, we invite you to submit a revised version of the manuscript that addresses the points raised during the review process.

The comments and suggestions are below.

We would appreciate receiving your revised manuscript by Jun 18 2020 11:59PM. To enhance the reproducibility of your results, we recommend that if applicable you deposit your laboratory protocols in protocols.io, where a protocol can be assigned its own identifier (DOI) such that it can be cited independently in the future. For instructions see: http://journals.plos.org/plosone/s/submission-guidelines#loc-laboratory-protocols

We look forward to receiving your revised manuscript.

Kind regards,

Fakir M Amirul Islam, PhD

Academic Editor

PLOS ONE

Additional Editor Comments (if provided):

A significant improvement, would like to suggest dealing some minor issues

Table 1: Data seems mean (95%CI), however, the title “Mean (SD)/ % (CI)” seems confusing. May be something could be presented referring a footnote.

Numbers are large, once decimal place would look better. I mean, instead of 66.38 (66.09 -66.66), 66.4 (66.1-66.7) throughout would look better.

Table 2: what is the difference between Model 1 and Model 2 (total sample). This is assumed, one is unadjusted and the other is multivariate-adjusted. This has not been reported. What covariates were in the Model is not known here?. * indicates significant, ** means more significant, *** very significant. In fact, all are significant. The confidence intervals automatically show which are significant and which are not. Therefore, the symbols are not necessary. I would suggest presenting like this,

Model 1 (total sample) IRR (95% CI)* Model 2 (total sample) IRR (95% CI)** Model 3 (only men) IRR (95% CI)**, Model 4 (only women) IRR (95% CI)**

*IRR(95%CI): Unadjusted model; **IRR (95%CI): adjusted for X,y,Z……………

If the authors are still interested to show the significance with a symbol, only one symbol could be shown and said this is for p<0.05.

Table 2: Age categories do not seem logical at all. 15-35, before adult (15-17), early adult (18-25), and adult (26-35) in one group. 30 to 35 or 36 to 45 do not seem any different in nature.

There could be references when there is an age group transition. Possibly, 15-25, 26-45, 46-60, 60+. I would consider some interesting findings could come up from this large sample. Could be checked for this variable only.

The variable, Overall job satisfaction (from 0 = worst to 10 = best). What does the non-significant protective association tell? As the satisfaction level increases by one point from 0 to 1 or 1 to 2, workplace absenteeism decreases a bit. This means, as the satisfaction level increases, less the absenteeism. There could have a threshold cut-off or binary cut-off where it could be significantly higher. I would suggest checking this variable if there is any binary cut-off where it could be significant

---

## [Author Response · Author response to Decision Letter 1]

5 May 2020

Please find the response in the 'Response to the Reviewers' file.

---

## [Editor Report · Decision Letter 2]

7 May 2020

Gender differences in the longitudinal association between obesity, and disability with workplace absenteeism in the Australian working population

PONE-D-19-32153R2

Dear Ms Keramat,

We are pleased to inform you that your manuscript has been judged scientifically suitable for publication and will be formally accepted for publication once it complies with all outstanding technical requirements.

With kind regards,

Fakir M Amirul Islam, PhD

Academic Editor

PLOS ONE

Additional Editor Comments (optional):

Congratulations!!
---

## [Editor Report · Acceptance letter]

11 May 2020

PONE-D-19-32153R2 

Gender differences in the longitudinal association between obesity, and disability with workplace absenteeism in the Australian working population 

Dear Dr. Keramat:

I am pleased to inform you that your manuscript has been deemed suitable for publication in PLOS ONE. Congratulations! Your manuscript is now with our production department. 

With kind regards,

on behalf of

Dr Fakir M Amirul Islam 

Academic Editor

PLOS ONE